# A Comparison of Clinical Efficiency of Photodynamic Therapy and Topical Corticosteroid in Treatment of Oral Lichen Planus: A Split-Mouth Randomised Controlled Study

**DOI:** 10.3390/jcm10163673

**Published:** 2021-08-19

**Authors:** Jacek Zborowski, Dorota Kida, Aleksandra Szarwaryn, Karol Nartowski, Patrycja Rak, Kamil Jurczyszyn, Tomasz Konopka

**Affiliations:** 1Department of Periodontology, Wroclaw Medical University, ul. Krakowska 26, 50-425 Wroclaw, Poland; tomasz.konopka@umed.wroc.pl; 2Department of Drug Form Technology, Wroclaw Medical University, Borowska 211 A, 50-556 Wroclaw, Poland; dorota.kida@umed.wroc.pl (D.K.); aleksandra.szarwaryn@umed.wroc.pl (A.S.); karol.nartowski@umed.wroc.pl (K.N.); patrycja.rak@umed.wroc.pl (P.R.); 3Department of Dental Surgery, Wroclaw Medical University, Krakowska 26, 50-425 Wroclaw, Poland; kamil.jurczyszyn@umed.wroc.pl

**Keywords:** oral lichen planus, photodynamic therapy, corticosteroid treatment, preclinical research, new concepts

## Abstract

Background: The aim of the study was to compare the effectiveness of photodynamic therapy (PDT) to steroid therapy in the treatment of oral lichen planus (OLP). Due to the lack of commercially available drug carriers, innovative proprietary solutions were used for both the photosensitiser and the steroid carrier—in the first case to shorten and in the second to extend the contact of the active substance with the mucosa. Methods: A prospective, randomised, single-blind, 12-week full contralateral split-mouth clinical trial of 30 patients with bilateral oral lichen planus was conducted. The prepared matrices were incorporated with active substances methylene blue 5% and 0,05% triamcinolone. The size of lesions, Thongprasom, ABISIS, and VAS scale were evaluated. Results: Relatively high rates of complete remission of lichen were demonstrated: immediately after treatment, 33.3% with PDT and 22.2% with triamcinolone (TA), and after 3 months, 54.2% with PDT and 62.9% with TA. After 3 months of treatment, a reduction in the area of evaluated lesions of 52.7% for PDT and 41.7% for TA was achieved. Conclusion: In situations of topical or general contraindications to oral corticosteroids, resistance to them, or the need for repeated treatment in a short period of time, PDT appears to be a very promising treatment option.

## 1. Introduction

Lichen planus is a chronic inflammatory mucocutaneous disease caused by an immune response to still not fully identified antigens. Cutaneous lesions of lichen planus are estimated to account for approximately 0.4 to 1.2% of all dermatological diagnoses [1], with two-thirds of these patients developing lichen planus on the oral mucosa, whereas in the mucosal variety, cutaneous lesions occur in only 16% of patients [2]. The global pooled prevalence of oral lichen planus (OLP) in a recent meta-analysis was 1.01%, with the highest prevalence in South-Central America at 1.74%, Europe at 1.32% and 0.47% in North America, as well as an almost 3.5 times higher prevalence in people over 40 years of age, compared to younger people [3]. Although the explanation of the aetiopathogenesis of lichen is not definitively conclusive, four main pathogenetic domains can be identified in this discussion. First, the attention is drawn to the immune dysregulation and activation of T cells, in particular, CD8^+^ cells destroying basal epidermal and/or epithelial keratinocytes, which is accompanied by hyperreactivity of the Th1 cells and secretion of pro-inflammatory mediators, e.g., interferon-gamma (INF-y) and proapoptotic mediators, e.g., caspase-3 and Fas ligand (sFASL) [1]. Proinflammatory signalling in OLP may also originate from Th17 cells [4]. In addition, the high tissue expression of chemokines (CLXCL9, CLXCL10, CLXCL11) indicates their key role in determining the inflammatory infiltrate in OLP, in particular T cell subpopulations [5]. Secondly, genetic susceptibility appears to be significant. In Caucasians, the occurrence of OLP has been shown to be associated with single nucleotide polymorphisms—tumour necrosis factor α (TNF-α), 308 Guanine/Adenie (G/A) for the GA genotype, and A allele and for IFN-γ UTR5644 for the TT genotype [6]. Thirdly, OLP is associated with viral infections—HCV, HPV 16 and 18, EBV and CMV, and fungal infections—*Candida* spp. (*albicans*, *tropicalis*, *parapsilosis*, *pintolopesii*), *Torulopsis glabrata*, *Alternaria* spp., and *Sclerotinia* spp. [7]. Finally, important risk factors for OLP are environmental factors leading to the formation of lichen-like lesions through type IV hypersensitivity (dental materials and especially amalgam, metals, and especially nickel sulphate) and oral lichenoid drug reactions to an increasingly long list of systemically used drugs [1,2,8]. These risk factors also include psychological factors in the form of stress, anxiety, and depression. Different types of OLP, particularly the red forms, lead to a significant reduction in oral health-related quality of life (OHRQoL) [9]. The worst complication of OLP is the possibility of malignant transformation of these lesions. A recent systematic review [10] and meta-analysis estimate this risk at 1.4 and 1.2%, respectively. The average time for this transformation is 61.9 months and most cases involve the lateral border of the tongue and red OLP variants [10], whereas the risk of malignancy is significantly increased in active nicotine users, frequent alcohol consumers, and those seropositive for HCV [11].

There is no uniform treatment protocol for OLP [2,8,12]. Cases of lichen-like lesions require medical background clarification whenever possible. Particularly in cases of suspected local contact allergies, simple measures to eradicate sensitising compounds, such as replacement of dental materials or correct removal of dental biofilm, can be effective. It is also necessary to eliminate the cause of frictional trauma of oral mucosa and all dental iatrogenies that can cause Köbner’s phenomenon. Treatment is not required for asymptomatic white variants of lichen [2,8]—the most common being reticular and limited plaques on the dorsal surface of the tongue. However, even these lesions require periodic checks because, as in most immune-mediated diseases, periodic exacerbations are common. The red forms—atrophic, erosive, and the rarest bullous—require absolute treatment. This is due to subjective complaints from the patient, and above all, due to the elimination of potential ‘cancerisation fields’ that predispose to the formation of malignant oral lesions [13].

Pharmacological treatment of OLP can be topical and/or general. Only mucosal cases of lichen are generally treated with topical administration of strong and very strong corticosteroids (acetonide triamcinolone, acetonide fluocinolone, fluocinonide, valerate betamethasone, propionate clobetasol), calcineurin inhibitors (cyclosporin, tacrolimus, pimecrolimus, sirolimus), and the least effective retinoids. Steroids remain the first clinical choice for the topical treatment of OLP [2,8,12,14]. Their anti-inflammatory effects related to inhibition of phospholipase A_2_ activity, inhibition of DNA and mRNA of pro-inflammatory factors, immunosuppressive effects on Langerhans cells and T cells, and shrinking and sealing effects on small blood vessels are utilised [15]. Previously, steroids were used in the form of nonadhesive or poorly adhesive mucosal ointments, liquid emulsions, gels, or tablets dissolved near lesions or even as solutions administered by submucosal injection. Currently, carriers of sodium carboxymethyl cellulose (Orabase) or hydroxyethyl cellulose or liposomal systems are used to ensure sufficiently long contact of the steroid with the moist mucosa. Relative to placebo, steroids in OLP result in a significantly better analgesic effect, whereas they do not result in better healing of inflammatory lesions and more adverse effects [14]. The improvement in all these clinical outcomes compared with calcineurin inhibitors remains controversial, although the issue of increased risk of malignant transformation after tacrolimus and pimecrolimus has recently been raised [8,14]. An important factor in the choice of topical steroid use in the treatment of OLP is only the extremely rare occurrence of iatrogenic Cushing’s syndrome or other systemic effects, and of topical complications, there is the possibility of iatrogenic candidiasis if the steroids are used for a prolonged period [2,8,12]. A nonselective effect of steroids in the pathogenesis of OLP is noted. General systemic pharmacological treatment of OLP is usually carried out in the mucocutaneous course of the disease or in the most resistant cases to topical treatment. In such cases, therapeutic options include the use of corticosteroids (prednisone, methylprednisolone) alone or in combination with immunosuppressive drugs (methotrexate, azathioprine, mycophenolate mofetil) or biologic agents (alefacept, efalizumab, basiliximab) [2,12].

The treatment of OLP also utilised diode laser ablation and before that, CO_2_ laser ablation. Low-level laser therapy (LLLT), by means of photobiomodulation, produced the following desired effects: anti-inflammatory, analgesic, and antiapoptotic, stimulating tissue regeneration and healing with minimal adverse reactions. A meta-analytic comparison of LLLT and topical corticosteroid therapy showed no significant differences in pain reduction or desired evolution of lesions [16]. However, this comparison involved small groups of treated patients with a short postoperative follow-up period. Diode lasers of different wavelengths (630–808 nm) were used and the established irradiation protocols varied widely.

Clinical trials for the use of photodynamic therapy in OLP have been ongoing since 2006 [17]. This treatment involves the penetration of a photosensitiser into over-proliferating inflammatory cells, including T cells (CD4^+^, CD8^+^, Th17), in the subepithelial connective tissue, followed by their selective apoptosis under the influence of oxygen free radicals formed after light beam emission, followed by epithelial healing with regeneration [18]. In addition, following PDT of oral lichen lesions, significant reductions in IL-17a^+^, CD4^+^CD137^+^, CD8^+^CD137^+^ T-cell counts, and CLXCL10 chemokine were observed in peripheral blood one month after treatment, also indicating a systemic anti-inflammatory effect [19]. To date, the results of studies comparing PDT with topical corticosteroid therapy in OLP in terms of pain reduction, reduction in lesion area and desired evolution of inflammatory lesions are divergent [18,19,20], as are the conclusions of two meta-analytical comparisons of PDT efficacy in this clinical application [16,21]. Limitations to the use of PDT in OLP are the disparate clinical management protocols, the variety of photosensitisers used, and the difficulty of maintaining them in the oral environment. 

The aim of this study was to evaluate the effectiveness of remission of inflammatory lesions of oral lichen planus under the influence of photodynamic therapy using methylene blue as a photosensitiser and urban steroid therapy after application of 0.5% triamcinolone acetonide, both substances incorporated into porous polymer matrices. In addition, the reduction of the lichenification lesion area and the evolution of the clinical picture were comparatively evaluated, as well as changes in patient oral health-related quality of life at follow-up 3 months after treatment. Innovative proprietary solutions were used for both the photosensitiser and the carrier for the steroid, serving in the first case to shorten and in the second to lengthen the contact of the active substance with the mucous membrane.

## 2. Materials and Methods

A prospective, randomised, single-blind, 12-week full contralateral split-mouth clinical trial of patients with bilateral erythematous or erosive oral lichen planus was conducted between October 2020 and April 2021. The selection of the split-mouth RCT method was motivated by a desire to reduce the effect of multiple variables on the efficacy of topical OLP treatment between patient groups. The respondents were recruited among those treated for OLP at the Outpatient Clinic for Mucosal and Periodontal Diseases of the Academic Dental Polyclinic of the Medical University of Wrocław. The study was approved by the Bioethics Committee of the Medical University of Wrocław (kb256/2020) and registered in clinicaltrials.gov with the number NCT04976673.

The needed sample size was calculated to compare proportions of two dependent samples (based on the McNemar test formula) assuming a type I error rate of 5%, the power as 0.8, the proportion of success in both groups as 0.75, the proportion of failure in both groups as 0.25. The sample size included in the study was 30.

The primary criterion for inclusion in the study group was a clinical finding of bilateral erythematous or erosive OLP, confirmed histopathologically after taking an excisional biopsy from the clinically worst site in the oral cavity. The histopathological examination was performed at the Department of Clinical Pathology, Medical University of Wrocław, and all specimens were evaluated by the same histopathologist. Modified WHO criteria for the clinical and histopathological diagnosis of OLP were adopted [22]. In addition, the histopathological examination assessed the severity of the inflammatory process on a semi-quantitative scale: + mediocre, ++ moderate, +++ significant. The exclusion criteria were dysplasia on histopathological examination, potential conditions for lichen-like lesions (diabetes, systemic use of drugs causing them, liver disease, graft-versus-host disease, local hypersensitivity type IV), nicotinism, pregnancy, and breastfeeding. An eligible patient had to have at least two bilateral OLP erythematous/ulcerative lesions greater than 1 cm in size requiring topical treatment. All eligible patients signed an acknowledgement that they understood the nature of the proposed treatment, consented, and were informed that they could discontinue treatment at any stage. Qualification into the study group along with the assignment of a number from 1 to 30 was carried out by the same doctor (TK).

Porous matrices were prepared by lyophilisation of foam obtained from dispersions of components mixed in appropriate quantitative ratios, respectively, pullulan (abcr, Karlsruhe, Germany), sodium salt of alginic acid (Sigma-Aldrich, Steinheim, Germany), methylcellulose (4000 cp, Sigma-Aldrich, Steinheim, Germany), and glycerol 95% (Sigma-Aldrich, Steinheim, Germany). Finally, the active substance triamcinolone acetonide (Carbosynth limited, Berkshire, UK) or methylene blue (abcr, Karlsruhe, Germany) was introduced into the homogeneous mixtures (Table 1). Foamed at 2340 rpm (Gako e/s, Eprus, Bielsko-Biala, Poland), the formulations of the mixtures were poured onto Teflon-coated Petri dishes, frozen at −26 °C and lyophilised at room temperature with an ultimate vacuum of 9.0 × 10^−2^ to 1.3 × 10^−1^ mBar (Lyovac GT2, Steris, Hurth, Germany) for 20 h.

The prepared matrices with incorporated active substances were evaluated for mechanical properties using a TA-XT plus texture analyser (Stable Micro System. Godalming, UK), the time of matrix smearing in an aqueous medium at 37 °C was determined, the pH of the extract formed after smearing was determined using the potentiometric method and the release of active substances over time from the prepared formulations into the water at 37 °C was carried out by taking samples for determination for triamcinolone acetonide by HPLC method and for methylene blue by UV–Vis method against prepared standard curves. The results of the physicochemical characterisation of the formulations with both triamcinolone acetonide and methylene blue are summarised in Table 2.

The developed dry porous polymer carriers with active substances, triamcinolone acetonide, and methylene blue, respectively, were characterised by physicochemical properties as favourable for application to oral mucosa in the treatment of inflammatory lesions. The matrices in the pharmaceutical evaluation were mechanically robust, resistant to smearing while releasing the active ingredients efficiently during in vitro studies.

The carrier was very precisely adjusted to the size of the mucosal lesion treated (Figure 1). Therefore, the concentration is given in mg/cm^2^ in a dry medium of the carrier.

On the one hand, the OLP lesion qualified for treatment was subjected to photodynamic therapy in four sessions every 2–3 days on days 1, 3, 6, 9. The lesion was completely covered with 5% methylene blue for 10 minutes, after which the lesion was irradiated with a diode laser (patent number: 9,223,123, B2; date of patent: 29 December 2015) with spot size 0.8 cm^2^ at 650 nm using energy fluence 120 J/cm^2^ and power density 1034 mW/cm^2^ for 227 s. The treatment was always carried out by the same doctor (KJ). The qualified OLP lesion, on the other hand, was treated by daily adherence of a carrier cut to the lesion size with 0.05% triamcinolone acetonide over a period of 9 days. On the days of PDT, a steroid preparation was applied to the lesion at the end of each session; on days 2, 4, 5, 7, and 8 of the treatment, a carrier adapted to the size of the lesion was self-administered by the patient after the last dental cleaning of the day. Both forms of treatment were carried out in parallel (Figure 2). The research leader (J.Z.) randomised the allocation of the OLP lesion side to the form of treatment as follows: patients with odd eligibility numbers on the right had PDT and on the opposite side steroid therapy, and patients with even numbers vice versa. 

Clinical evaluation of the evolution of OLP lesions was conducted at baseline, at the end of both forms of treatment (day 9), and after 12 weeks without treatment. In relation to the treated sites, the following were used: modified OLP treatment efficacy scale by Carrozzo and Gandolfo [23] (complete remission—withdrawal of the lesion or its transition to a white lesion; partial response—marked reduction of the erythematous or erosive lesion area; no response—no marked reduction of the red lesion area or its deterioration); evaluation of the clinical lesion area using a PCP UNC15 periodontometer (Hu-Friedy, Chicago, IL, USA) scaled in 1 mm increments (the greatest lesion length and width were measured and the area in mm^2^ was calculated on this basis), and the OLP lesion evolution scale under treatment according to Thongprasom et al. [24]. The following were assessed per patient: time in years since the diagnosis of OLP in the oral cavity, number of teeth on both sides of the oral cavity, autoimmune bullous skin disorder intensity score (ABSIS) [25] (ABSIS1—for assessing the extent of skin lesions from 0 to 150 points; ABSIS2—for assessing the extent of oral lesions from 0 to 11; ABSIS3—for assessing discomfort while eating or drinking from 0 to 45 points), visual analogue scale (VAS from 0 to 10), and oral-health realised quality of life (OHRQoL) using a shortened version of the oral health impact profile (OHIP-14) index [26]. Parameters relating to treated sites and VAS were assessed at all three follow-up visits; other variables were assessed at baseline and after 12 weeks. At each follow-up visit, any adverse effects of the ongoing treatment were also reported, both in relation to the oral cavity and to clinically detectable general changes. Clinical assessment was conducted by the same doctor unaware of the randomisation method (T.K.).

To assess the normality of the distribution of the variables, the Shapiro–Wilk test was applied. In the statistical analysis, the Student’s *t*-test for dependent variables (used only for the assessment of OHIP-14 changes after treatment) or the Wilcoxon paired rank test was used to assess differences between variables. The McNemary test was used to assess differences between frequencies of binary scale variables. Spearman’s test was used in correlation analysis. To assess the effect of independent variables on the improvement in the Thongprasom et al. scale, multiple regression models were created. The models were verified on the basis of: the significance of partial regression coefficients, the absence of collinearity between independent variables, the existence of homoscedasticity, the absence of autocorrelation of the residuals (Durbin–Watson test), the normality of the distribution of the residuals, and the 0 value of the random component ε_i_. To assess the effect of independent variables on the remission of OLP or lack of it, according to the modified Carrozzo and Gandolfo scale after 3 months of follow-up, a logistic regression model was proposed and verified by the significance of the regression function, the significance of the regression coefficients, the normality of the distribution of the residuals and the mean value of the residuals equal to 0. For each test except correlation, *p* < 0.05 was considered statistically significant; in the analysis of covariation, the significance threshold was *p* < 0.02. Statistica 13.3 software was used for statistical analysis.

## 3. Results

Of the 30 OLP patients eligible for treatment, 4 discontinued the PDT and 1 the topical corticosteroid therapy without discontinuing treatment on the other side, while 2 patients did not attend the final examination. Finally, the results of OLP treatment were analysed in 28 patients, and with regard to lesions, 24 were treated with PDT and 27 with TA, as shown in the flow diagram (Figure 3).

Table 3 summarises the demographics and baseline clinical parameters for the entire treated group of subjects and those treated with the two protocols for bilateral red oral lichen lesions. Older women with mostly long-standing and multifocal OLP predominated strongly in the analysed group. Histopathological examination indicated the presence of an increased inflammatory process in the subepithelial connective tissue. Seven patients had a mucocutaneous form of lichen that did not require general treatment. The occurrence of OLP-related pain and its intensity varied significantly.

This includes the link between the oral condition of treated people with lichen and quality of life. The greatest impact on worsening of OHRQoL in OLP was noted for the physical pain (3.61 ± 2.8) and psychological discomfort (3.29 ± 2.3) domains. A total of 30 erythematous or erosive lesions on the buccal mucosa, 15 on the gingiva (exfoliative gingivitis), and 6 on the tongue (generalised plaque or plaque rubella) were treated. There were no significant differences in the analysed clinical parameters for the lesions qualified for treatment on both sides of the oral cavity, except for a larger area of the lesions treated with PDT. However, there was a significant correlation in the area of OLP lesions between oral sites (R = 0.65, *p* = 0.003).

Table 4 summarises the efficacy results of the two OLP treatment protocols by altering the modified scale by Carrozzo and Gandolfo. Relatively high rates of complete remission of lichen were demonstrated: immediately after treatment, 33.3% with PDT and 22.2% with TA, and after 3 months, 54.2% with PDT and 62.9% with TA. On each occasion, a significantly more frequent complete remission of OLP, as described by Carrozzo and Gandolfo, was found after both forms of therapy, with a significantly stronger statistical basis for rejecting the hypothesis of only slight or no clinical improvement three months after both therapies.

The values of changes in the area and the Thongprasom scale of OLP lesions treated with both methods are shown in Table 5. Only three of the analysed lesions (5.8%) showed complete clinical remission and all of them concerned the erythematous gingival form. In relation to the baseline study, a significant decrease in lesion area following PDT and topical steroids was found in both of the other observations (Figure 4 and Figure 5). After 3 months of treatment, a reduction in the area of evaluated lesions of 52.7% for PDT and 41.7% for TA was achieved. The loss of the erythematous component of the OLP lesions and their transition to white reticular forms is shown by Thongprasom scale change analysis. While lesions still remained erythematous in the study immediately after treatment, further significant clinical improvement occurred at the 12-week clinical follow-up and was better noted for topical corticosteroid therapy.

The administered topical treatment did not significantly affect the cutaneous lesions when they cooccurred with mucosal lesions (Table 3). Three months after treatment, significantly fewer OLP (ABSIS2) occupied oral sites were observed. The treatment resulted in a significant reduction in the patient’s oral pain. This was demonstrated for three independent parameters (ABSIS3 three months after treatment, VAS immediately after treatment and three months later, and physical pain domain in OHIP-14 three months after treatment). In contrast, the conducted treatment was not found to significantly improve the other OHRQoL domains or the OHIP-14 summary score (Table 6).

There were no significant covariations between pre- and post-treatment clinical variables between patients (data not shown). Only completely obvious positive correlations were found between VAS and OHIP-14 before treatment (R = 0.56, *p* = 0.001) and three months after treatment (R = 0.62, *p* < 0.000). A positive correlation was also observed between the Thongprasom scale and OHIP-14 three months after treatment (R = 0.6, *p* < 0.000).

The lack of bivariate relationships creates opportunities for multivariate regression models. For improvement of the Thongprasom scale (from 4 to 0—no improvement or worsening), the most statistically significant multiple regression model indicated a significant effect of age and baseline lesion area, with no significant effect of treatment type. The multiple regression equation has the following form:

Thongprasom scale change = 0.03 ∗ age-0.02 ∗ output surface area of the lesion—0.197 ± 0.966.

Model verification data: F = 5.28 and *p* = 0.008; R = 0.42; R^2^ = 0.18; partial regression coefficients for age 0.35 and *p* = 0.013, and for surface area −0.34 and *p* = 0.015; no collinearity between independent variables—tolerance 0.94; homoscedasticity based on a residual plot against predicted values; uniform scatter plot, the autocorrelation of residuals- d in Durbin–Watson test 1.62 and R = 0.18; distribution of residuals based on scatter plot of residuals against normal value; normal distribution; random component ε_I_; mean Cook distance 0.02.

For OLP remission according to the modified Carrozzo and Gandolfo scale at final follow-up, the combination of age, baseline area, and the number of years with oral lichen had the strongest effect in the logistic probability regression model, with no significant effect of the protocol of the undertaken treatment. The logit regression model has the following formula:

Logit *p* = −3.665 + 0.092 ∗ age-0.052 ∗ exit surface area of the lesion—0.047 ∗ years with OLP.

Model validation data: Chi^2^ = 10.54 and *p* = 0.014; OR for age 1.10 (1.02–1.18), for surface area 0.95 (0.9–0.0997) and for years with OLP 0.86 (0.74–1.01); significance of regression coefficients: age *p* = 0.012, surface area *p* = 0.04, years with OLP *p* = 0.069, distribution of residuals based on scatter plot of residuals against expected normal value—normal distribution, mean value of residuals 0.01.

Topical side effects of the therapy were observed only during the 9-day active treatment period. Four patients experienced an exacerbation of OLP inflammatory lesions, slight swelling, and increased pain after the first or second PDT session, resulting in disagreement with continuing therapy on this side of the mouth. One elderly patient abandoned self-administration of the polymer carrier with TA as a result of technical problems with drug insertion. One patient reported exacerbation of halitosis in relation to treatment. No general adverse effects were observed during the treatment period and in the post-treatment follow-up. There was no established reason for the failure of two treated patients to attend the last follow-up visit.

## 4. Discussion

The innovative use of mucosa-adhesive porous polymer matrices as carriers for MB and TA is a desirable solution to many clinical difficulties. In previous comparative studies of PDT with MB or toluidine blue (TB) in relation to topical OLP steroid therapy, photosensitisers have been administered directly to the lesion using a micropipette [27] to the oral rinse [28,29,30,31], or by soaking sterile swab and applying them simultaneously to symmetrical lesions [32]. Such varied methods of photosensitiser application introduced a significant interference factor to clinical observations and, in practice, made it impossible to obtain adequate concentrations in the affected tissue, e.g., on the tongue. The clinical efficacy of PDT in oral pathologies is highly dependent on the photosensitiser used, which among other things, should have low dark toxicity and high photocytotoxicity, good selectivity towards target cells and rapid elimination, good absorption and solubility in biological fluids, and the ability to bind to carrier systems without losing its biological properties [33]. In comparative studies similar to our own, the mode of topical steroid application also varied—a whole-mouth rinsing with dexamethasone [27,28,31] or as an adherent TA [29,32] and betamethasone valerate [30]. The first way of administration, in particular, offered the possibility of developing iatrogenic candidiasis and increased the likelihood of systemic adverse effects. The use of an adhesive polymer matrix with TA allowed selective and prolonged steroid action, while lamination of the outer surface reduced side effects. Additionally, the use of proprietary polymer matrices allowed for split-mouth RCTs to be conducted by eliminating spill-over effects, which also seems to be innovative for oral pathology [34,35] Split-mouth design was only used in single observations comparing the clinical effectiveness of two OLP treatments [36,37].

Multifocal OLP appears suitable for the use of a split-mouth trial in the evaluation of within-person intervention studies in which there is random allocation of symmetrical lesions on both sides of the mouth to the treatment strategy being evaluated. This reduces the impact of many unknown aetiological determinants with respect to the model of randomisation of individuals into parallel groups. Lesions are treated at similar levels during the same disease exacerbation, thus reducing the period effect, which is important in the dynamic progression of oral lichen. Two levels of evaluation of treatment efficacy are obtained: unilateral lesions treated with PDT or topical steroid (evaluation according to the modified Carrozzo and Gandolfo scale, Thongprasom scale, and surface area) and the level of qualified patients treated with bilateral topical therapy (ABSIS, VAS, and OHIP-14) [35]. In addition, the use of a split-mouth trial requires a smaller number of patients to discover the same treatment effects with respect to parallel-group design, although recruitment itself is more difficult with the former model. Parallel RCTs have been used to compare the efficacy of PDT and topical steroid therapy in OLP [27,28,29,30,31]. Only one study [32] used a concurrent timing of interventions on both sides of the mouth, but on the control side, one of the options was laser irradiation of lesions without photosensitiser, and six sequences of therapy were randomised. Taking into account these methodological differences, comparisons of our results to other authors’ observations on the efficacy of these OLP treatments in parallel groups should be treated with great caution.

With respect to our own observations, all other similar comparative studies [27,28,29,30,31] were conducted in subjects who were on average at least a decade younger. Only one study [28] did not show a significant female predominance among those treated for OLP. In two studies [27,31], the mean duration of OLP was on average more than 10 years longer. In one case [31], histopathological examination was not used as a diagnostic criterion for OLP; in another [28], the reticular variety was also treated. Two clinical observations [29,30] concerned only the erosive form of OLP. A distinctive feature of our own study is that we also undertook the treatment of the erythematous gingival form of OLP, in the form of desquamative gingivitis always coexisting with histopathologically verified other variants of OLP. The inclusion of this form of lichen only increased the difficulty of follow-up and was a response to the common symptomatology of such a form among our patients. The higher age of our patients was due to the lack of an upper age limit in the inclusion criteria and may not have included all cases of drug-induced lichen-like lesions in the elderly as a result of a failure to disclose a history of all generally used drugs.

The main outcome of the study was to assess the remission of erythematous and erosive OLP lesions according to the Carrozzo and Candolfo scale. Complete remission on this scale means a transition to reticular OLP or absence of any lesion on follow-up examination. The latter eventuality in our patients three months after treatment occurred for only three treated gingival lesions (5.8% of OLP lesions, in two after steroid, and in one after PDT). For such a defined remission, only Mostafa et al. [29] presented better results 2 months after treatment, with 28.9% of cured lesions—seven with PDT and four after steroid. Still, only one further observation [30] in 12.5% of patients 1 month after PDT treatment showed complete remission of treated lichen foci. In the two earliest studies [27,28], such evolution of lesions was not found any case. Our own results defined by the evolution of the lesions to reticular variation or complete withdrawal are auspicious. Attention is drawn to the increase in such remissions during the three-month follow-up period by 21% with PDT and about 40% of the lesions with the steroid. Only slightly worse results 6 weeks after PDT treatment alone were reported by Cosgarea et al. [19]—35% of patients with complete remission, 60% with partial remission, and 5% with no improvement. The observations of other authors are clearly worse: after 2 months, complete remission according to the Carrozzo and Gandolfo scale after PDT treatment in 36.8% and after steroid treatment in 21% of lesions [29], one month after PDT treatment in 0% and 12.5% of patients, and after steroid treatment in 14.2% and 25% of patients [27,30]. All these clinical observations, including our own, show no significant difference in achieving complete remission of OLP between PDT and topical steroid therapy. For our study, the Carrozzo and Gandolfo scale [23], which in the original describes the evolution of only erosive OLP lesions, was slightly modified. Since erythematous lesions predominated in our observations, it was proposed that a partial response should be defined as a marked reduction in the area of OLP red lesions, and no response as an imperceptible reduction in their area or a worsening of the clinical picture. Recurrence of OLP during the 3-month follow-up period was demonstrated in 11.8% of the lesions treated in our centre (three after PDT treatment and three after steroid treatment in each of the treated sites). Mirza et al. [31] documented the recurrence of OLP 12 months after treatment in 10% of patients, only following PDT. However, a five-year clinical follow-up indicates a significantly higher 35.7% recurrence rate of red OLP forms in patients treated with topical steroids [38]. This indicates a strong need to seek alternative treatments for OLP.

The secondary outcome was the measurement of changes in the surface area of OLP lesions following treatment. Different methods have been used thus far digital calliper [31], graded tongue blade [28,32], and transfer of lesion size to sterile flexible transparent paper and then to millimetre paper [29]. In our study, we used a simple and fairly estimative method to assess the OLP lesion surface area, as described by Sobaniec et al. [39]. The PDT-induced reduction in lichenification surface area of 52.7% and 41.7% after steroid treatment in our patients corresponds to the values reported by other authors—57.1% and 38.3%, respectively [30] and 44.3% 3 months after PDT treatment [17]. Additionally, the analysis of favourable changes in the Thongprasom scale under the influence of both forms of treatment confirms the frequent transition of the red lesions into white lesions, which still occurred after the application of the steroid in the polymer matrix during the three-month follow-up period. The obtained mean reduction of 1.16 in the Thongprasom scale after PDT treatment and 1.52 after steroid treatment differs from the observations of other authors—after 7 weeks the mean value of this scale was reduced by 1.87 after PDT and 1.12 after steroid treatment [32] and after 2 months by as much as 3.16 after PDT and only 1.0 after steroid treatment [29]. These differences may be due to the advancement of the initial clinical forms of lichen—in the erosive form, very significant clinical improvement is obtained after PDT, but less so in the erythematous form, especially of the gingiva, which weighed in our evaluation. A meta-analytic evaluation of the reduction in Thongprasom score under these two forms of therapy combining four observations [27,28,29,32] and involving a total of 53 lesions treated with PDT and 56 with steroid showed no significant difference with statistically significant heterogeneity in the included study results [21]. Analysis of the surface area changes and the evolution of OLP lesions in Thongprasom scale under the influence of treatment show that these lesions only decrease and it is possible to obtain a white keratinised reticular form, which, however, may be the starting point of the next exacerbation of the disease.

In our own observation, mucocutaneous lichen occurred in 25% of patients. The mean value of the cutaneous ABSIS1 index was very low and did not change significantly following local OLP treatment. In multivariate analyses, the cooccurrence of skin lesions was not observed to significantly affect the clinical efficacy of the ongoing therapy. Cosgarea et al. [19] with a similar baseline value and lower standard deviation of ABSIS1 at 5 and 6 weeks after PDT showed a significant reduction in the value of this index. However, it does not seem possible that four sessions of oral PDT can significantly affect the healing of skin lesions, which does not eliminate further interest in this issue. Both the self-observation 3 months after the end of treatment and the comparative study [19] demonstrated a significant reduction in the number of anatomical locations in the oral cavity occupied by OLP lesions already one week after the end of PDT. This indicates the possibility of withdrawal of lichen lesions in areas that were not treated, thus having a carryover effect, in our observation remote in time.

Assessment of the analgesic effect is inherent in any study of the efficacy of OLP treatment and the most commonly used index for this purpose is the visual-analogue scale and its modifications [14,18,40]. The ongoing bilateral topical treatment among our patients resulted in a significant reduction in pain intensity especially immediately after the therapy. This effect was independently confirmed by the index of discomfort when eating or drinking (ABSIS3). However, this indicator does not appear to be particularly useful in OLP, where only specific foods or drinks cause discomfort, and many patients identify a well-defined and highly variable dietary component causing the symptom. Deciding which therapy method leads to a better analgesic effect is possible in parallel-group design studies. The very first study evaluating PDT in OLP indicated a complete absence of pain immediately after this therapy [17]. Mostafa et al. [29], two months after treatment of erosive OLP, showed a mean VAS reduction of 7.3 versus 2.9 after steroid treatment. A meta-analysis of six applications of PDT with different photosensitisers in the treatment of 88 OLP lesions indicated that the mean VAS reduction was 3.82, with significant heterogeneity among the included studies [21]. This study indicated that diode laser as a light source and MB and TB as photosensitisers were most effective in abolishing pain. Topically applied steroids in OLP result in better pain control with respect to placebo, a better analgesic effect is produced by mucosally applied steroids, and no advantage in abolishing pain intensity has been proven for the selected topical corticosteroid compound [14]. At the level of a meta-analysis pooling 4 studies [27,29,31,32], 53 patients showed no significant difference in analgesic effect between PDT and topical steroids, with significant heterogeneity in these observations [21].

OLP significantly worsened OHRQoL based on the OHIP-14 index in a meta-analysis of 4 studies in 222 patients versus 563 healthy oral mucosa subjects, although the difference was only 0.85 and the mean OHIP-14 determined from 21 observations was 15.2 with a confidence interval of 12.2–18.2 [9]. Daumme et al. [41] showed significantly higher mean OHIP-14 values in red OLP forms (15.6 ± 10.9), compared to white forms, with significantly higher mean for three domains—physical pain, physical disability, and psychological discomfort. The baseline OHIP-14 in our patients was therefore within this limit. Additionally, although the ongoing treatment resulted in an improvement in this index by an average of 2.54, this change was not statistically significant. This improvement was essentially due to a significant reduction in the perception of pain intensity, with no significant treatment effect on the other six domains. No comparative study of PDT versus topical steroid in OLP has assessed OHRQoL. At 6 weeks after PDT in OLP, a significant improvement in QoL was described (index name not given), and this was due solely to a statistically significant decrease in burning [19]. A significant reduction in OHIP-14 (mean change of up to 11.9) was observed after 6 weeks of oral rinsing with 0.4% TA, but 44% of those treated developed candidiasis [42]. In contrast, multi-month follow-up of patients treated with topical TA showed a negative effect on the full version of the OHIP index and on 7 of its domains [43]. The heterogeneity of all these observations is probably due to the wide variance in aetiological conditions, the dynamic variation in clinical forms and subjective symptoms of OLP, and the different timing of postoperative evaluation. 

The possible side effects after topical corticosteroid therapy are well known, although not witnessed in our observation. Four PDT patients (13.3%) at our centre developed an inflammatory oedematous reaction after the first or second treatment, which discouraged them from continuing this treatment. Some OLP patients treated with PDT have previously described mild burning sensation on the day of treatment and persistent discomfort for several days [17,29,44]. This is due to the formation of reactive oxygen species, nitric oxide immediately after irradiation, as well as the release of pro-inflammatory cytokines and histamine and a neurogenic mechanism related to the activation of TRP receptors [45]. There is insufficient evidence to consider the LED light profile as safer in this regard compared to the laser beam, whereas modulation of power density during radiation may provide a solution to this clinical problem [45]. In one study, 40% of those treated with PDT for OLP one day after treatment were described to have slight oedema [29]. This may be a form of phototoxic inflammation associated with histamine release [45]. Despite these possible complications, PDT in the treatment of lichen is considered a safe method [18,20,21,40], which can be an alternative to topical corticosteroid therapy also due to its many topical and general side effects.

To select the variables associated with the success of the conducted therapy, two regression models were created to evaluate the dependent variable in an ordinal scale (evolution of the Thongprasom scale) and a dichotomous variable (obtaining or not complete remission of the lesion according to the Carrozzo and Gandolfo scale). The chosen explanatory variables significantly affecting these parameters were the patient’s age (the higher the age is, the greater extent of favourable evolution of the Thongprasom scale and 10% higher probability of OLP remission for each year are observed), the initial lesion surface area (the smaller the surface area is, the greater the extent of favourable evolution of the Thongprasom scale and 5% higher probability of remission of the inflammatory form of OLP for each cm^2^ of the less extensive lesion are observed) and OLP history (the shorter the lesion persists in the oral cavity, the higher the probability of remission of the disease is). The interpretation of the relationship with the initial efflorescence surface area is most obvious. Immune reactivity decreases in the elderly and perhaps this should be explained by the greater possibility of red OLP lesions being eradicated in them. Conversely, the persistence of OLP lesions increases the possibility of their resistance to the treatment, which may be related to increased chromatin condensation and functional acetylation of histone H3 (Lys9) in the basal layer of the epithelium [46]. Additionally, clinically, OLP treatment with PDT has been shown to have better results in shorter-lived lesions versus more persistent lesions [47]. The inclusion of treatment type in the models did not improve regression significance. This is further evidence of the comparable effect of both therapies on the clinical improvement exponents of red OLP lesions. Additionally, such a result, although clinically unsatisfactory, in the light of the possibility of evaluation of lesions to white after PDT, can be interpreted as a search for a viable alternative to the first choice in OLP treatment and in line with oral oncological prevention.

## 5. Conclusions

The conducted research has a number of limitations. Firstly, we did not have clear diagnostic criteria for lichen-like lesions, and their exclusion was only based on medical history. This could have been a confounding factor for the whole observation, especially in the case of polypragmasy. Secondly, the choice of a split-mouth model prevented the assessment of the placebo effect and double blinding, and did not eliminate carryover effects inside and between the sides of the mouth. Thirdly, a clinically simple but nevertheless quite estimative method was used to assess the area of the OLP lesions. Certainly, analysis of the structural changes of lichen foci under treatment in a different light on repeated intraoral radiographs may provide a solution to this clinical limitation. Finally, there are no clear guidelines on the optimal PDT protocol in OLP. We believe that our empirical choice of the light source, photosensitiser and form of administration, number of treatments, and also radiation power and energy dose is appropriate, but this requires confirmation in an in vitro study.

In addition to our own study, comparative and methodologically correct assessments by other authors [29,30,32], systematic reviews of the literature [18,20,40] and a meta-analysis [21] provide the basis for sanctioning photodynamic therapy as a second-line method in the treatment of OLP. In situations of topical or general contraindications to oral corticosteroids, resistance to them, or the need for repeated treatment in a short period of time, it appears to be a very promising treatment option. We hope that our proprietary way of applying photosensitiser will be accepted by a wide range of clinicians, as it finally meets the condition of bioabsorbability for different locations of oral lesions. In the latest European guidelines on OLP therapy choices [48], photodynamic therapy is not even mentioned; our proposal is that it should at least be included in second-line treatments.

## Figures and Tables

**Figure 1 jcm-10-03673-f001:**
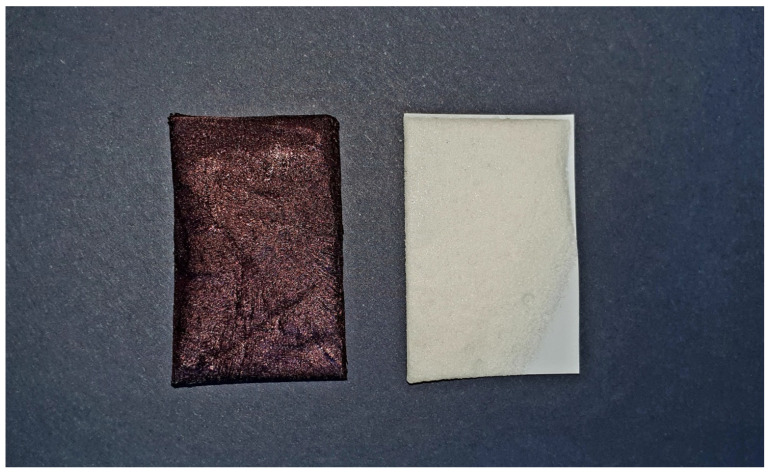
Carrier with methylene blue (**left**) and triamcinolone acetonide (**right**) before application.

**Figure 2 jcm-10-03673-f002:**
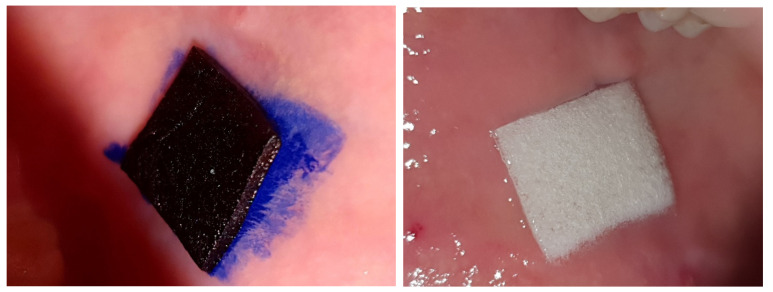
Carrier with methylene blue (**left**) and triamcinolone acetonide (**right**) cut to the lesion size and applied on the mucosa.

**Figure 3 jcm-10-03673-f003:**
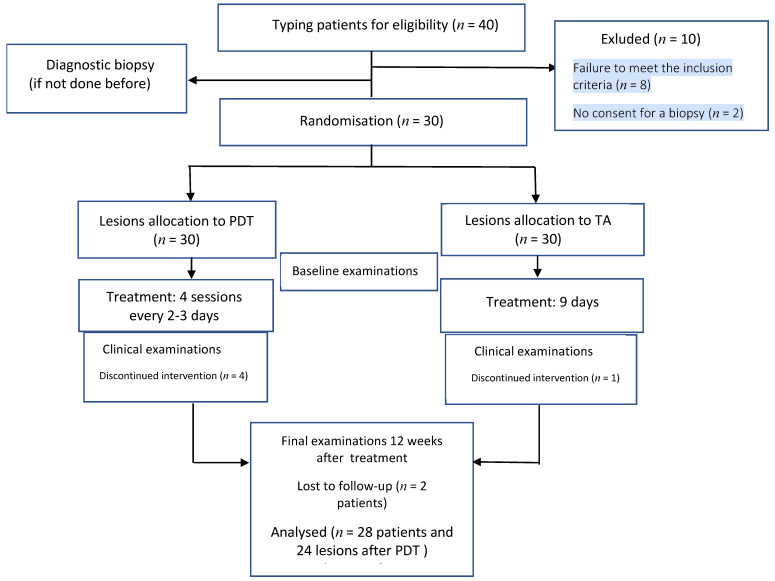
Participant and sites flow, treatment, and assessment period during 3 months.

**Figure 4 jcm-10-03673-f004:**
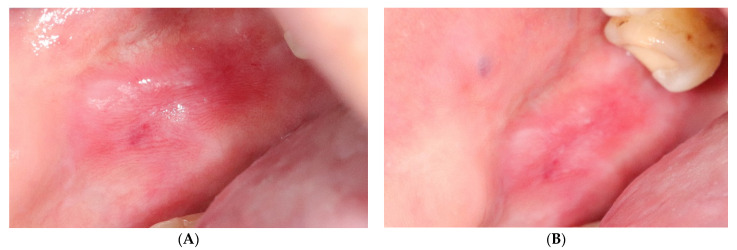
Oral lichen planus before (**A**) and after (**B**) triamcinolone therapy.

**Figure 5 jcm-10-03673-f005:**
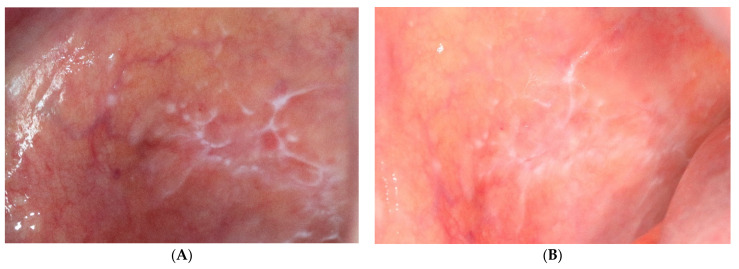
Oral lichen planus before (**A**) and after (**B**) photodynamic therapy.

**Table 1 jcm-10-03673-t001:** Composition of porous matrices with triamcinolone acetonide (TA) and methylene blue (MB). Evaluation of the physicochemical properties of the prepared matrices.

Formulation Code	Composition (% in Dry Mass)				
	Pullulan	Sodium Alginate	Glycerol	Methylcellulose	Active Substance
TA	36.6	9.8	48.7	4.85	0.05
MB	27.0	7.3	38.9	4.3	22.5

**Table 2 jcm-10-03673-t002:** Porous matrix with incorporated drug characteristics. Data are presented as the mean ± SE, *n* = 3.

Formulation Code	Paremeters				
	Elongation (%)	Rupture Force (g)	pH	Disintegration Time (min)	Cumulative % Release of Drug
TA	189.10 ± 28.00	253.20 ± 34.60	6.85	>300	49.23 ± 5.25 after 150 min of release test
MB	142.91 ± 6.53	1498.70 ± 325.94	6.61	>300	79.24 ± 11.30 after 195 min of release test

**Table 3 jcm-10-03673-t003:** The baseline demographic and clinical data of enrolled patients and sites with OLP.

Variables	Patients (28)	Treatment Lesions PDT (24) Steroid (27)	
Age: interval, mean (SD)	33–76, 61.9 (10.9)	62.3 (11.1)	61.7 (11.2)
Gender: F/M (*n*)	24/4	20/4	23/4
Intensity of inflammation in histopathological examination (*n*)	+ 2 ++ 12 +++ 14		
OLP duration in years: interval, mean (SD)	1–20, 5.21 (4.8)	5.8 (5.0)	4.5 (4.0)
Number of teeth: median	4–31, 21.5	10.5	11
ABSIS1: No > 0, mean (SD)	7, 0–27, 3.88 (7.6)		
ABSIS2: interval, median	2–7, 3		
ABSIS3: No > 0, mean (SD)	21, 0–33, 6.27 (8.1)		
VAS: interval, mean (SD)	1–10, 4.86 (2.7)		
OHIP-14: interval, mean (SD)	0–33, 13.68 (9.2)		
Location of treated OLP lesion		Buccal mucosa: 14	16
		Gingiva: 6	9
		Tongue: 4	2
Lesion size in cm^2^: mean (SD)		17.8 (16.6)	15.5 (13.2)
Thongprasom score: interval, median, score (*n*)		2–5, 3	2–5, 3
		Score 2-5	Score 2-1
		Score 3-17	Score 3-23
		Score 4-1	Score 4-1
		Score 5-1	Score 5-2

+ mediocre, ++ moderate, +++ significant.

**Table 4 jcm-10-03673-t004:** Evaluation of treatment results according to modified Carrozzo and Gandolfo scores.

Type of Treatment	Immediately after Treatment CR PR No Response			3 Months after Treatment CR PR No Response		
PDT	8 ^A^	10	6	13 ^C^	5	6
TA	6 ^B^	18	3	17 ^D^	5	5

CR complete response, PR partial response. ^A^ *p* = 0.013; ^B^ *p* = 0.041; ^C^ *p* = 0.009; ^D^ *p* = 0.0001.

**Table 5 jcm-10-03673-t005:** Changes in means lesion size and Thongprasom score for OLP lesions in both treatment methods.

Lesions Clinical Scores	Treatment Method	^1^ Baseline	^2^ Immediately after the End of Treatment	^3^ After 3 Months Follow-Up	*p* Values
Lesion size in cm^2^	PDT	17.83 ± 16.6	10.71 ± 11.7	8.43 ± 6.9	1 vs. 2 and 1 vs. 3 *p* < 0.000 2 vs. 3 *p* = 0.51
	TA	15.53 ± 13.2	9.12 ± 8.4	9.06 ± 10.3	1 vs. 2 and 1 vs. 3 *p* < 0.000 2 vs. 3 *p* = 0.43
Thongprasom score (means and medians)	PDT	2.87 ± 0.7, 3	2.13 ± 1.0, 2	1.71 ± 1.0, 1	1 vs. 2 *p* = 0.005, 1 vs. 3 *p* = 0.004, 2 vs. 3 *p* = 0.17
	TA	3.15 ± 0.6, 3	2.41 ± 1.0, 3	1.63 ± 1.1, 1	1 vs. 2 *p* = 0.002, 1 vs. 3 *p*< 0.000, 2 vs. 3 *p* = 0.004

^1^ Baseline, ^2^ Immediately after, ^3^ After 3 Months.

**Table 6 jcm-10-03673-t006:** Changes in clinical variables and oral health-related quality of life for OLP patients due to the treatment taken.

Variables in Patients	^1^ Baseline	^2^ Immediately after the End of Treatment	^3^ After 3 Months Follow-Up	*p* Values
ABSIS1 (No > 0, means)	7, 3.88 ± 7.6	ND	7, 2.41 ± 4.7	1 vs. 3 *p* = 0.1
ABSIS2 (median)	3	3	3	1 vs. 2 *p* = 0.06, 1 vs. 3 *p* = 0.036, 2 vs. 3 *p* = 0.08
ABSIS 3 (means)	6.27 ± 8.1	ND	3.91 ± 6.6	1 vs. 3 *p* = 0.02
VAS (means)	4.86 ± 2.7	1.96 ± 2.5	2.86 ± 2.6	1 vs. 2 *p* < 0.000, 1 vs. 3 *p* = 0.002, 2 vs. 3 *p* = 0.13
Overall OHIP-14	13.68 ± 9.2	ND	11.14 ± 9.0	1 vs. 3 *p* = 0.14
Functional limitations	1.04 ± 1.6	ND	0.46 ± 1.2	1 vs. 3 *p* = 0.11
Physical pain	3.61 ± 2.8	ND	2.75 ± 2.5	1 vs. 3 *p* = 0.044
Psychological discomfort	3.29 ± 2.3	ND	2.79 ± 1.9	1 vs. 3 *p* = 0.08
Physical disability	1.5 ± 2.1	ND	1.25 ± 1.8	1 vs. 3 *p* = 0.64
Psychological disability	1.75 ± 1.8	ND	1.07 ± 1.3	1 vs. 3 *p* = 0.13
Social disability	1.57 ± 1.6	ND	1.86 ± 2.0	1 vs. 3 *p* = 0.41
Handicap	0.93 ± 1.2	ND	0.93 ± 1.2	1 vs. 3 *p* = 0.96

^1^ Baseline, ^2^ Immediately after, ^3^ After 3 Months.

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
