# Peer review of "A Comparison of Clinical Efficiency of Photodynamic Therapy and Topical Corticosteroid in Treatment of Oral Lichen Planus: A Split-Mouth Randomised Controlled Study"

_jcm, 2021, doi:10.3390/jcm10163673_

Round 1

Reviewer 1 Report

Firstly: Thank you for this Research 

anybody performing OLP studies knows about the Burden and difficulties

M&M: aqueous solution for release….. just water? Saline Solution?

saliva Would be interesting

PDT was performed on day 1,3,6,9? 
3 or 4 times?

Plaque Index was Not assessed?

The Discussion is very Long and Some paragraphs like Changes in Surface are difficult to read; might shorten and review?!

Author Response

Comments and Suggestions for Authors

Firstly: Thank you for this Research 

anybody performing OLP studies knows about the Burden and difficulties

We are pleased that you appreciated the effort put into our research. Thank you very much for your comments and I hope that after introducing changes, our work will meet your expectations

M&M: aqueous solution for release….. just water? Saline Solution?

saliva Would be interesting

In vitro triamcinolone acetonide release studies

Samples of the triamcinolone acetonide loaded matrix with a size of 25 × 25 mm with closely similar mass were immersed in 15 mL water and incubated at 37 °C in a sealed closed vessel and wrapped in aluminum foil with horizontal shaking 60 times per minute. Accurately 1.5 mL of each sample was withdrawn at predetermined time intervals (30, 60, 90, 120, 150, 180, 210, 240, 270, 300, 330, and 360 min) filtered through 0.45 micron membrane filter and analysed to determine the of drug concentration in samples using HPLC. These solutions were analysed using Agilent 1260 Infinity Quaternary System HPLC (Agilent Technologies Ltd., Stockport, UK) connected to UV/Visible spectrophotometer which was set at 240 nm

In vitro methylene blue release studies

The fragments of polymer matrices (2.5 x 2.5 cm) were inserted in three 50 mL orange volumetric vessel, then 10 mL of distilled water was added. The samples were placed in shaking water bath in temperature of 37 °C, set at with horizontal shaking 60 times per minute. The samples were protected with parafilm and aluminium foil against the water evaporation and sunlight. he samples to methylene blue determinations were taken at 16 time points to 195 minutes. From each of three vessels 200 µL samples was taken and water diluted to 625-fold finally. After samples taken, to all vessels was added equivalent water volume with the temperature of 37 °C. Based on calibration curve, the amount of release methylene blue in time were calculated.

Due to the restriction of work on human material due to Covid in the laboratory, in the process of designing carriers, we used methods that reflected the water environment of the oral cavity.

During the study the carrier was dissolved and had adhesion to the mucosa thanks to saliva. It was applied in a dry form.

PDT was performed on day 1,3,6,9? 
3 or 4 times?

PDT was performer 4 times on 1, 3 , 6 , 9 day

This was also made more precise after your remark in the text

Plaque Index was Not assessed?

PI was not assesed during the study but all the patients participating in the program were under the care of the periodontology clinic, so the hygiene was high and negligible as a problem of the study

The Discussion is very Long and Some paragraphs like Changes in Surface are difficult to read; might shorten and review?!

Thank you for your suggestion. I tried to make some changes to the discussion. It has been shortened where possible. I hope it will be more readable now

Reviewer 2 Report

 Comparison of Clinical Efficiency of Photodynamic Therapy and

Topical Corticosteroid in Treatment of Oral Lichen Planus- Split-Mouth

Randomised Controlled Study

Journal: Journal of Clinical Medicine

General comments 

This is a well-organized clinical research. Clearly the results do not differ from the results of similar studies for oral lichen planus.

However, there are no well-organized clinical trials for the treatment of oral lichen planus, so any information is valuable.

I suggest publishing after minor corrections.

Minor comments 

What does TA  in line 21 (abstract ) stand for?

INF-y and pro-apoptotic mediators, e.g. caspase-3 and 43 sFASL (1). line 43 please define sFASL and INF-y

 lines 48-49 …: OLP has been shown to be associated with single nucleotide polymorphisms- TNF-α -308 G/A for the GA genotype and A allele and for IFN-γ UTR5644 for the TT genotype (6) (Please define the acronyms)

LINE 89 e (Orbase) … probably refers to ORABASE

line 100 General pharmacological …. probably refers to systematic

please comment on the risk of malignant transformation due to the prolonged use of steroids as referred previously: " The treatment of choice, mainly topical corticosteroids (LozadaNur, 2000; Gonzalez-Moles et al, 2002, 2003; Gonza´lezMoles and Scully, 2005a,b), may, it has been proposed, make patients more vulnerable to malignant transformation (Duffey et al, 1996) " in "Oral lichen planus: controversies surrounding malignant transformation" by MA Gonzalez-Moles, C Scully, 2008, Oral DiseasesVolume 14, Issue 3 p. 229-243) JA Gil-Montoya 

Note :figures 4and 5 do not show a significant difference in clinical outcome and are not uniform in parameters (exposure, distance, color). If the patients had a significant improvement in pain that needs to be clarified.

Please comment,  How do you explain the continuous effects of treatment after the cessation of treatment?

Could diode laser have a prolonged effect in Oral mucosa immunocytes? Same could happen with topical steroids?

Author Response

General comments 

This is a well-organized clinical research. Clearly the results do not differ from the results of similar studies for oral lichen planus.

Thank you kindly.

However, there are no well-organized clinical trials for the treatment of oral lichen planus, so any information is valuable.

I suggest publishing after minor corrections.

Minor comments 

What does TA  in line 21 (abstract ) stand for?

It stands for triamcinolone acetonide. The change was made to the text

INF-y and pro-apoptotic mediators, e.g. caspase-3 and 43 sFASL (1). line 43 please define sFASL and INF-y

sFASL  Fas ligand (FasL, a ligand of the Fas cell death receptor) is a key factor in the regulation of these processes. FasL is primarily found in two forms: full length (membrane, or mFasL) and cleaved (soluble, or sFasL).

INF  Interferon‐gamma (IFN‐γ) is a cytokine that plays an important role in inducing and modulating an array of immune responses. Cellular responses to IFN‐γ are mediated by its heterodimeric cell‐surface receptor (IFN‐γR), which activates downstream signal transduction cascades, ultimately leading to the regulation of gene expression.

 lines 48-49 …: OLP has been shown to be associated with single nucleotide polymorphisms- TNF-α -308 G/A for the GA genotype and A allele and for IFN-γ UTR5644 for the TT genotype (6) (Please define the acronyms)

TNF-α - tumor necrosis factor α

G- Guanine

A- Adenine

TNF-α and TNF-β genes are located in tandem on chromosome 6 between the Class I and Class II cluster of the major histocompatibility complex (chromosome 6p21.1–6p21.3). TNF-α-308G/Apolymorphism (rs1800629) has been reported to be associated with several autoimmune/inflammatory diseases including OLP. The genetic variation at position −308 of the TNF-α gene results in two allelic forms in which the presence of guanine (G) defines the common variant and the presence of adenine (A) defines the less common one.

Interferon‐gamma : IFN-γ gene at position UTR5644 (A/T) Adenine /Tyrosine

The frequencies of allele A and genotype GA of TNF-α (-308G/A) are significantly higher while allele G and GG genotypes were lower in OLP

Allele A (TNF-α 2 allele) lies on the extended haplotype HLA-A1-B8-DR3-DQ2, which is associated with high TNF-α production.

The allele A (TNF-α 2 allele) has been demonstrated to be a much stronger transcriptional activator than the common allele G (TNF-α 1 allele).

The changes were made to the text.

LINE 89 e (Orbase) … probably refers to ORABASE

Yes it does. The change was made to the text.

line 100 General pharmacological …. probably refers to systematic

Yes it does. a change was made to the text.

Please comment on the risk of malignant transformation due to the prolonged use of steroids as referred previously: " The treatment of choice, mainly topical corticosteroids (LozadaNur, 2000; Gonzalez-Moles et al, 2002, 2003; Gonza´lezMoles and Scully, 2005a,b), may, it has been proposed, make patients more vulnerable to malignant transformation (Duffey et al, 1996) " in "Oral lichen planus: controversies surrounding malignant transformation" by MA Gonzalez-Moles, C Scully, 2008, Oral DiseasesVolume 14, Issue 3 p. 229-243) JA Gil-Montoya 

Theoretically, topical corticosteroids could be carcinogenic in some patients due to their immunosuppressive effects. On the other hand, their anti-inflammatory action reduces the risk of neoplastic transformation. Recent systematic reviews of the literature on the risk of skin cancer development as a result of long-term topical use of steroids ended with negative recommendations in this regard: Ratib S, Burden-Teh E, Leonardi-Bee J, Harwood C, Bath-Hextall F. Long-term topical corticosteroid use and risk of skin cancer: a systematic review. JBI Database System Rev Implement Rep. 2018,16(6):1387-1397 – „We did not find any studies that might help establish if long-term TCS use is associated with skin cancer”. Also, the observation of the topical use of corticosteroids in OLP did not confirm an increased risk of such transformation: Otero-Rey E.M, Suarez-Alen F, Peñamaria-Mallon M, Lopez-Lopez J, Blanco-Carrion A. Malignant transformation of oral lichen planus by a chronic inflammatory process. Use of topical corticosteroids to prevent this progression? Acta Odontol Scand. 2014: 72:8, 570-577 – “The treatment with topical corticosteroids has an anti-inflammatory and immunosuppressive effect. The maintenance of this therapy may inhibit the chronic inflammatory processes present in OLP and in this way prevent the progression to malignant transformation.”. Additionally, in our own study, cell dysplasia was excluded in the histopathological examination before starting the treatment, which fully protected against this possible side effect. The treatment protocol we adopted included only a 9-day local steroid therapy.

Note :figures 4and 5 do not show a significant difference in clinical outcome and are not uniform in parameters (exposure, distance, color). If the patients had a significant improvement in pain that needs to be clarified.

All photographs were taken using Canon EOS 77D, Canon 60 mm f/2.8 EF-S USM Macro lens  (Canon, Ōta, Tokyo, Japan) with Metz 15 MS-1 ring light (Metz, Markham, ON, Canada). All images were taken from the same distance (45 cm) with the optical axis of the camera kept perpendicular to the surface of the lesion. The focus point was locked to 45 cm to achieve repeatability.

Only 3 of the analysed lesions (5.8%) showed complete clinical remission and all of them concerned the erythematous gingival form. In relation to the baseline study, a significant decrease in lesion area following PDT and topical steroid was found in both of the other observations (Fig. 4 and Fig. 5). After 3 months of treatment, a reduction in the area of evaluated lesions of 52.7% for PDT and 41.7% for TA was achieved. You can find reduction of red component of OLP on picture but still it is not complete reemission as noticed in the beginning.

Please comment,  How do you explain the continuous effects of treatment after the cessation of treatment?

Thank you very much for this interesting and difficult question. The posteroid effect of OLP treatment is mainly to inhibit the inflammatory response and its duration is unpredictable. Gonzales-Moles et al. (Oral Dis. 2018; 24(4): 573-579) they rightly distinguished two types of patients with OLP, characterized by a completely different effectiveness of local steroid therapy, for whom they proposed a different treatment protocol. Unfortunately, so far there are no clear criteria for the selection of the optimal treatment, replacing the empirical choices of the treatment method, its intensity and duration. In this situation, the persistence of the steroid effect in OLP is random and essentially depends on the selection of patients included in the observation (e.g. age, polypharmacy, stress, genetically determined immune reactivity, duration of oral lesions). Certainly, the technology of steroid administration is of significant importance for the improvement of the effectiveness of OLP treatment. We believe that the innovative mucoadhesive method of steroid administration proposed in our study contributed to a relatively good treatment result, described within only 12 weeks.

Further clinical observation of these patients indicates that the effect was however worsening. The beneficial effect of PDT on OLP is also explained by the immunomodulatory effect, consisting in selective apoptosis or necrosis of excessively proliferating lymphocytes T. Additionally, it is possible to use PDT in the case of OLPs that are OPMDs (Jin X, Xu H, Deng J, Dan H, Ji P, Chen Q, Zeng X. Photodynamic therapy for oral potentially malignant disorders. Photodiagnosis Photodyn Ther. 2019;28:146-152) and in the case of OLP superinfection Candida spp. Also in the case of this treatment, there are mechanisms of its ineffectiveness, about which less is known due to the still small number of studies compared to topical steroid therapy in OLP. These can be the mechanisms mentioned above, as it is also a form of symptomatic treatment, they can also be protocol-related considerations (type of photosensitizer, wavelength, spot size, energy fluence and power density). Our clinical experience, however, shows a greater possibility of controlling variables related to the PDT protocol itself, especially after the use of proprietary matrices with incorporated methylene blue. The need to standardize the PDT protocol for OLP management is obvious or to select it for the yet unknown clinical variables.

Could diode laser have a prolonged effect in Oral mucosa immunocytes? Same could happen with topical steroids?

In the available literature, there are few results of studies on changes in the number of inflammatory cells in the infiltrate after OLP treatment. Cosgarea et al. Photodynamic therapy in oral lichen planus: A prospective case-controlled pilot study. Sci Rep. 2020; 10: 1667 two weeks after the application of PDT with a diode laser and methylene blue at a concentration of 10 mg / mL showed a significant reduction in the number of CD4 + and CD8 + cells in the inflammatory infiltrate. The presence of plasmocytes in the inflammatory infiltrate of OLP is associated with a lower likelihood of recurrence and a better response to topical corticosteroid therapy. (Anitua E, Piñas L, Alkhraisat MH. Histopathological features of oral lichen planus and its response to corticosteroid therapy: A retrospective study. Medicine (Baltimore). 2019;98(51):e18321. doi:10.1097/MD.0000000000018321). This subpopulation of cells can be stimulated when treating erosive forms of OLP with PRGF (Piñas L, Alkhraisat MH, Suárez-Fernández R, Anitua E. Biomolecules in the treatment of lichen planus refractory to corticosteroid therapy: Clinical and histopathological assessment. Ann Anat. 2018;216:159-163). There are no studies showing a longer effect in OLP remission after PDT with respect to steroids. Such studies in conjunction with the immunohistochemical assessment of changes in the inflammatory infiltration would be extremely interesting.